# Mix-Ecom: Towards Mixed-Type E-Commerce Dialogues with Complex Domain Rules

**Chenyu Zhou**[1,2,*]**, Xiaoming Shi**[*]**, Hui Qiu**[2]**, Yankai Jiang, Shaoguo Liu**[2]
**Tingting Gao**[2]**, Haitao Leng**[2,†]**, Xiawu Zheng**[1,†]**, Rongrong Ji**[1]

[1]Key Laboratory of Multimedia Trusted Perception and Efficient Computing,
Ministry of Education of China, Xiamen University
[2]Kuaishou Technology

## Abstract

E-commerce agents contribute greatly to helping users complete their e-commerce needs. To promote further research and application of e-commerce agents, benchmarking frameworks are introduced for evaluating LLM agents in the e-commerce domain. Despite the progress, current benchmarks lack evaluating agents' capability to handle mixed-type e-commerce dialogue and complex domain rules. To address the issue, this work first introduces a novel corpus, termed Mix-ECom, which is constructed based on real-world customer-service dialogues with post-processing to remove user privacy and add CoT process. Specifically, Mix-ECom contains 4,799 samples with multiply dialogue types in each e-commerce dialogue, covering four dialogue types (QA, recommendation, task-oriented dialogue, and chit-chat), three e-commerce task types (pre-sales, logistics, after-sales), and 82 e-commerce rules. Furthermore, this work build baselines on Mix-Ecom and propose a dynamic framework to further improve the performance. Results show that current e-commerce agents lack sufficient capabilities to handle e-commerce dialogues, due to the hallucination cased by complex domain rules. The dataset will be publicly available.

## 1 Introduction

**Large language models** (LLMs) (Meta, 2024; Jiang et al., 2024; Team et al., 2024; Ouyang et al., 2022; OpenAI, 2023; Brown et al., 2020) have revolutionized the backbone of agents for various application scenarios, such as medicine (Li et al., 2024; Mishra et al., 2024), finance (Zhang et al., 2024; Axtell & Farmer, 2025), education (Zhou, 2025; Xu et al., 2024), e-commerce (Nie et al., 2024; Zeng et al., 2025), etc. Among these domains, e-commerce agents attract increasing attention due to their attractive application value (Palen-Michel et al., 2024; Wang et al., 2025b; Pokrywka et al., 2025; Li et al., 2025a;b).

In the e-commerce domain, LLM-based agents contribute greatly to helping users complete their specific e-commerce needs, including real-world customer issues in the process of the pre-sale, logistics, and after-sale. To promote further research and application of e-commerce agents, several benchmarks have be proposed, as shown in Table 1. Current benchmarks evaluate e-commerce agents' performance mainly on simplistic user issues and e-commerce rules, which lack an objective and fair evaluation of agents' performance in real e-commerce scenarios. As shown in Figure 1, in real-world e-commerce dialogues, e-commerce rules are complex, and real-world user needs are dynamically changing, which requires agents to accurately understand complex domain rules and meet diverse user needs in one dialogue (the ability to handle mixed-type dialogues).

To address the issue, this work first introduces a new **mix**ed-type **e-com**merce customer-service dialogue dataset (Mix-ECom) for evaluating e-commerce agents' capability in real-world e-commerce dialogues. Specifically, high-quality mixed-type dialogues are first selected from 70,000 real world-service dialogues, which contains 4,799 dialogues, covering multiple dialogue types (task-oriented

---

[*]Equal Contribution
[†]Corresponding Author

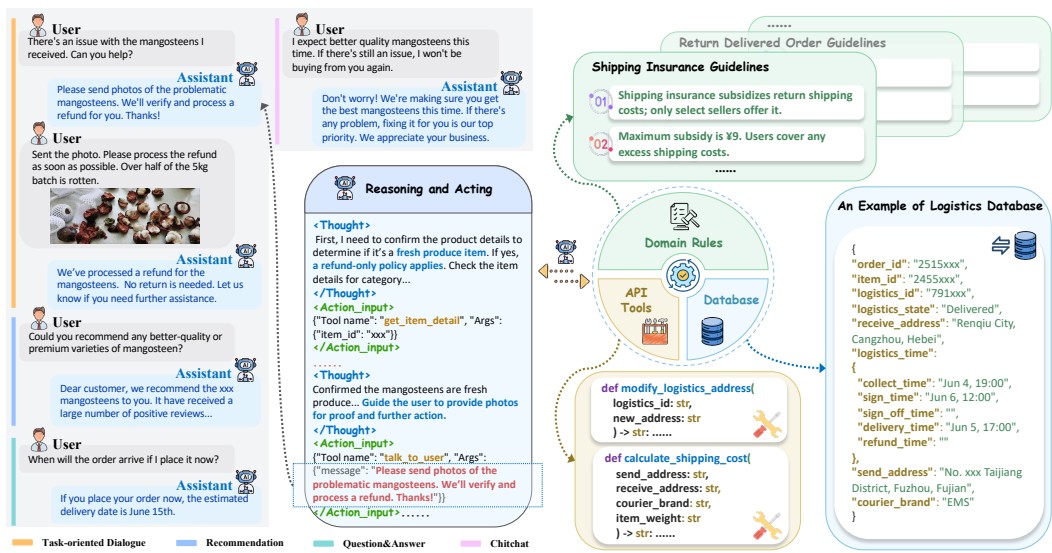

Figure 1: An example in Mix-ECom. The assistant needs verify the complaint based on the image provided and then take appropriate follow-up actions accordingly with given domain rules, database, and tools.

dialogue, recommendation, QA, chitchat) in one dialogue together, with 82 domain rules. Given the dialogues, the relative domain rules, API tools, and logistics database are also provide for further research, as shown in Figure 1. Furthermore, to ensure the high quality of the dataset, post-processing of privacy-preserving, chain-of-thought adding, manual filtering are conducted.

To analyze current agents' performance on Mix-ECom, 4 closed-source LLMs and 1 open-source LLMs are utilized as the backbone of agents under the setting of 0-shot. Besides, for further analysis, the open-source LLM is further fine-tuned with e-commerce dataset. Extensive experiments find that current agents lack sufficient capabilities to handle e-commerce dialogues, due to the hallucination caused by complex domain rules. To address this issue, this work proposes the dynamic e-commerce framework, which utilizes a dynamic module to select closely-related rules and reduce disturb from irrelevant rules, thus improving performance.

This work makes the following contributions:

- For an objective and fair evaluation of agents' performance in real e-commerce scenarios, we propose a novel benchmarking with a real-world mixed-type e-commerce dialogue dataset, termed Mix-Ecom, in which each session has rich variability of dialogue types and user intents, with 82 domain rules, API tools, and logistics database.

- To promote further research on e-commerce agents, we build baselines on Mix-ECom and propose a dynamic e-commerce framework to improve the performance on complex e-commerce rules.

- Experimental results show current agents lack sufficient capabilities to handle e-commerce dialogues, due to the hallucination caused by complex domain rules. Besides, the results show the effectiveness of the proposed dynamic e-commerce framework.

## 2 RELATED WORK

**LLM-based Agents.** Research on intelligent agents powered by LLMs represents a significant frontier in artificial intelligence. The open-source community has contributed several influential frameworks, including ReAct (Yao et al., 2023), Plan and Solve (Wang et al., 2023), LangChain (Chase, 2022), and AutoGPT (Gravitas, 2023), which provide foundational architectures for agent development. Besides, domain-specific enhancements have been achieved through specialized tool integration.

| Benchmark | E-com Task Types | | | Image | Video | # of Rule | Mixed-type |
|-----------|-----|------|-------|-------|-------|-----------|------------|
| | Pre | Logi | After | | | | |
| EcomScriptBench (Wang et al., 2025c) | ✗ | ✗ | ✗ | ✓ | ✗ | - | ✗ |
| CBYS (Zeng et al., 2025) | ✗ | ✗ | ✗ | ✗ | ✗ | - | ✗ |
| RECBENCH-MD (Liu et al., 2025) | ✗ | ✗ | ✗ | ✓ | ✓ | - | ✗ |
| Tau-retail (Yao et al., 2024) | ✓ | ✗ | ✗ | ✗ | ✗ | 32 | ✗ |
| ECom-Bench (Wang et al., 2025a) | ✓ | ✓ | ✓ | ✗ | ✓ | - | ✗ |
| Mix-ECom-Bench (Ours) | ✓ | ✓ | ✓ | ✓ | ✓ | 82 | ✓* |

Table 1: Comparison of e-commerce benchmarks. "-" indicates that the metric is not publicly available. "Pre", "Logi", and "After" represents pre-sales, logistics, and after-sales, respectively. "*" represents four dialogue types: QA, recommendation, task-oriented dialogue, and chitchat.

Search capabilities have been advanced through systems like WebGPT (Nakano et al., 2021) and WebCPM (Qin et al., 2023), while RestGPT Song et al. (2023) has demonstrated the potential of combining LLMs with RESTful APIs for web service development. However, there has been relatively little research focused on the domain of e-commerce customer service. To address the specific challenges in this scenario, we propose a novel dynamic e-commerce agent framework.

**Benchmark for E-commerce Agents.** Current benchmarks for evaluating e-commerce agents are increasingly evolving from uni-modal to multi-modal and multi-task settings. However, most existing benchmarks still offer only partial coverage of the e-commerce domain. For instance, EcomScriptBench (Wang et al., 2025c) supports scripted dialogue generation with both text and image inputs, but does not incorporate video. CBYS (Zeng et al., 2025) is confined to text-only inputs and is limited to product question-answering tasks. RECBENCH-MD (Liu et al., 2025) focuses primarily on product recommendation. Although Tau-retail (Yao et al., 2024) introduces the concept of domain-specific policy, its policy representations are overly simplified and deviate significantly from those used in real-world applications. To address these limitations, we propose Mix-ECom-Bench, a comprehensive benchmark that spans the full spectrum of e-commerce tasks and incorporates both image and video modalities with complex domain rules.

## 3 DATASET CONSTRUCTION

### 3.1 DATASET FORMULATION

The mixed-type e-commerce dialogue generation aims to generate responses $\mathcal{R}$ based on multi-modal files $\mathcal{F}$ (images sent by the customer, product detail images from the knowledge base, and recent live-streaming clips related to the product), user queries $\mathcal{Q}$, domain rules $\mathcal{P}$, tools $\mathcal{T}$.

In the dataset, each sample contains seven items, represented as $\{v, \tau, \alpha, o, \delta, \kappa, \theta\}$. The user profile $v = \{u^a, u^d\}$ includes the basic information of the current customer $u^a$ and their specific demands $u^d$, which are used to guide the LLM in simulating both customer and agent interactions. $\tau$ represents the reference plan to resolve the problem of customer, along with the reasoning process used to generate the plan. The action chain $\alpha = \{a_i\}_{i=1}^M$, each action is $a_i = \{a_i^m, a_i^p\}$, where $a_i^m$ is the tool name used in $action_i$ and $a_i^p$ and $o$ indicate arguments and action response. $\kappa$ represents the key answers the assistant must convey to the customer in this task. $\theta$ denotes the question type derived from the user profile $v$. Finally, $\delta$ represents the database information, which is not directly visible to the assistant but can be accessed indirectly through the tool set $\mathcal{T}$.

### 3.2 DATA CONSTRUCTION

#### 3.2.1 DOMAIN RULES, DATABASE AND TOOLS

The domain rules $\mathcal{P}$ explicitly defines the rules that the assistant must follow. It is derived from real-world e-commerce customer service practices and comprises 82 detailed rules, posing a significant challenge to the instruction-following capability of model. Further details are provided in the Appendix B.1. The database $\mathcal{D}$ is a sanitized version of real business data, stored in JSON format,

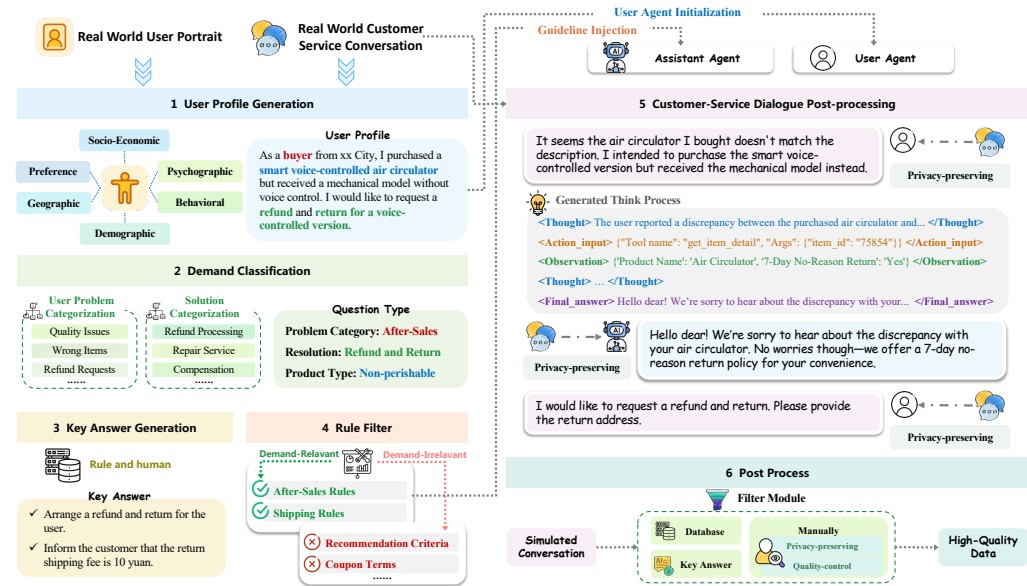

Figure 2: The construction process of Mix-ECom. The user profile, user demands, key answers, relative rules are first extracted from real world user portraits and customer-service conversations. Then, the dialogues are revised with post-processing of privacy-preserving and CoT adding. Finally, the dataset is reviewed manually to ensure high-quality.

and includes logistics, order, product, merchant, and user databases. The contents of the database are not directly visible to the user or assistant agents; they can only interact with it through predefined Tools. The tools $\mathcal{T}$ are manually written Python functions categorized into read, write, and converse operations, as detailed in the Appendix C.

### 3.2.2 DATA CONSTRUCTION PIPLINE

**User profile generation.** As illustrated in Fig 2, 70,000 real-world customer-service conversations are collected, along with real world user portraits derived from historical behaviors of users. GPT-4o is employed to summarize this information and generate corresponding user profiles. For logistics-related tasks, where customer requests are relatively straightforward, multiple user needs are combined to increase the complexity of the tasks. In after-sales tasks, images sent by users in actual customer service dialogues are preserved as multi-modal files $\mathcal{F}$. These are provided by the user agent during the roll-out process to validate the authenticity of their complaints. For pre-sales tasks, product detail images and recent live-stream clips are stored as multi-modal files $\mathcal{F}$. The model is required to extract relevant information from these files to answer product-related questions.

**Demand Classification.** Each task is being classified into specific question types $\theta$ through analysis of the User Profile, together with the status of products, orders, and logistics. Further details regarding question type generation are provided in Appendix B.2.

**Key Answer Generation.** The key answer ($\kappa$) and the ground truth database are being generated based on the identified question type ($\theta$) and predefined rules. The key answer is the information that must be conveyed to the user by the assistant agent.

**Rule Filter.** Due to the complexity of the complete Domain Rules $\mathcal{P}$, they are categorized into multiple sub-rules based on their content. For each task, filtering has been performed according to the question type $\theta$ of the task, retaining only the rules relevant to the current task. This rule filtering mechanism has significantly reduced hallucinations in the Assistant Agent and thereby improved the usability rate of the generated data.

**Customer-Service Dialogue Post-processing.** The user profile is being utilized as a prompt to simulate the customer through DeepSeek-R1, thereby constituting the user agent. Concurrently, the

assistant agent is constructed with GPT-4o serving as the backbone and the ReAct framework guiding its architecture. Real-world customer service dialogues are being employed as input for both the user agent and the assistant agent. These agents are being directed to rewrite the dialogue content, filter out privacy-sensitive information, and—in the case of the assistant agent—reconstruct the reasoning process in ReAct format based on responses originally provided by human assistants.

The agent receives a query enclosed by `<Question>` and `</Question>`. It then generates a reasoning process enclosed by `<Thought>` and `</Thought>`, followed by a tool call enclosed by `<Action_input>` and `</Action_input>`. After interacting with the environment through the tool, the agent receives the result enclosed by `<Observation>` and `</Observation>`. Once sufficient information is gathered, the agent produces a final response enclosed by `<Final_Answer>` and `</Final_Answer>`, marking the end of the thinking process.

### 3.3 DATA QUALITY CONTROL

A three-stage pipeline is employed for curating the e-commerce dialogue corpus.

**Manual Filtering of User Profiles and Question Types.** Manual filtering is being performed based on the obtained user profiles and question types to remove substandard user profiles as well as data with incorrect question type annotations.

**GPT-4o Filtering.** During the rollout phase, conversation content from interactions between the Assistant Agent and the User Agent, along with the resulting database state, is being collected. GPT-4o is then being utilized to evaluate two aspects: first, whether all Key Answers are present in the dialogue; and second, whether the database matches the ground truth database.

**Manual Meticulous Filtering.** A final review of flagged conversations is being conducted by human evaluators, who are rectifying omissions of key answers and filtering out dialogues containing sensitive user information, violations of fundamental e-commerce reasoning (e.g., unrealistic pricing, inconsistent policies), or any form of user privacy data.

Starting from an initial set of 70,000 data samples, the filtering process ultimately yields 299 test samples and 4,500 high-quality training instances.

### 3.4 DATA STATISTICS

Tasks within the benchmark dataset are classified into three categories: logistics, pre-sales, and after-sales. Logistics tasks are defined as handling all customer inquiries and requests associated with the shipping process. Pre-sales tasks concentrate on product discovery and persuasive communication, with a focus on purchase facilitation. After-sales tasks deal with post-purchase interactions, specifically addressing customer complaints and product returns.

**Statistics.** Mix-ECom, as shown in Table 2, contains 4,500 training and 299 test instances. Logistics tasks write to the database in 53.7 % of cases and invoke 6.66 tools on average. Pre-sales tasks are multi-modal—70 % include images and 30 % videos—but never write; they need 6.39 calls. After-sales tasks embed images in 76.9 %, and demand the most calls (7.37). Overall, 46.7 % of tasks contain images, 10 % contain videos, 41.7 % write to the database, and the mean number of tool calls is 6.79.

**Data Quality Evaluation.** 100 instances are selected from the test set and evaluated by five professional e-commerce customer service staff using a 0–1 scoring scale, where score 0 indicates low quality, and score 1 indicates high quality. To quantify the inter-annotator agreement, Fleiss Kappa was applied. The Fleiss Kappa coefficient is 0.76, indicating substantial agreement among annotators. Additionally, 86% of the data samples receive a score of 1, indicating a high level of quality and reliability in the dataset.

## 4 DYNAMIC E-COMMERCE AGENT FRAMEWORK

In e-commerce customer service, tasks differ sharply from traditional agent benchmarks in two key ways. First, domain rules are complex: e-commerce relies on an intricate set of policies (up to 82 in our dataset), placing strict demands on the instruction-following abilities of model. Second, user

|  | # of Test | # of Training | w/ Image | w/ Video | Write Db. | AVG. Tool Calls |
|---|---|---|---|---|---|---|
| Logistcs | 108 | 1,500 | 0.0% | 0.0% | 53.7% | 6.66 |
| Pre-sales | 100 | 1,500 | 70.0% | 30.0% | 0.0% | 6.39 |
| After-sales | 91 | 1,500 | 76.9% | 0.0% | 70.3% | 7.37 |
| Total | 299 | 4,500 | 46.7% | 10.0% | 41.7% | 6.79 |

Table 2: Basic statistics of Mix-ECom. "Write Db." represents requiring writing to the database. "Avg. Tool Calls" represents the average number of tool calls needed to resolve a task.

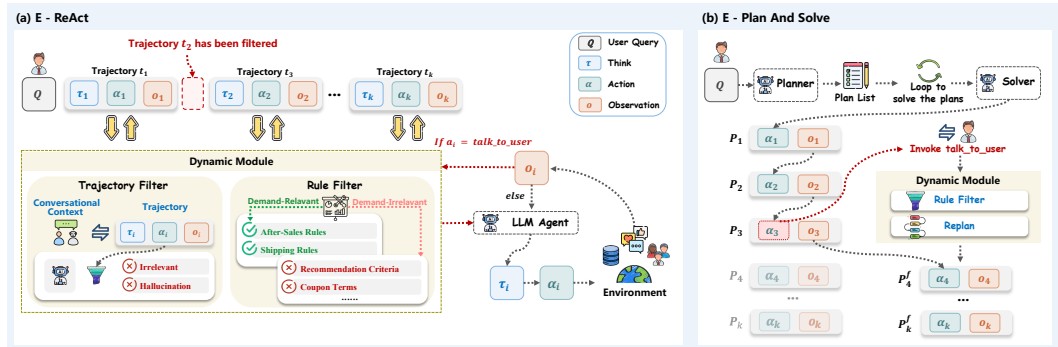

Figure 3: The illustration of the dynamic e-commerce agent framework, including E-ReAct and E-Plan&Solve, which are ReAct and Plan&Solve agent framework fused with the dynamic module to handle complex domain rules. E-ReAct in the block (a) filters the trajectory and the related domain rules, pruning irrelevant information to mitigate hallucination in subsequent reasoning. E-Plan&Solve in the block (b) utilizes the dynamic module to filter the domain rules and re-plans the remaining sub-tasks, enabling the system to meet users' changing needs.

queries are ambiguous—often vague or incomplete, and heavily dependent on context like product info, order status, and logistics data—requiring multi-turn interactions for models to infer user intent accurately. To tackle these challenges, we propose a dynamic e-commerce customer service module: E-ReAct and E-Plan&Solve.

## 4.1 E-REACT

E-ReAct is derived by tailoring the ReAct Yao et al. (2023) framework to the e-commerce domain. As illustrated in Figure 3, at step $t$ the vanilla ReAct agent receives the tuple $\{\mathcal{F}, \mathcal{Q}, \mathcal{P}, \mathcal{T}, \mathcal{H}_t\}$ where $\mathcal{P}$ denotes the Domain policy, and $\mathcal{H}_t$ the reasoning trajectory accumulated over the preceding $t$ steps,

$$\mathcal{H}_t = (\tau_0, \alpha_0, o_0, \tau_1, ..., \tau_{t-1}, \alpha_{t-1}, o_{t-1}).$$

Here, $\tau_i$ denotes the thinking process at step $i$, $\alpha_i$ stands for the action taken at step $i$, $o_i$ represents the feedback resulting from the action at step $i$.

To mitigate the dual challenges of policy complexity and query ambiguity, we introduce a dynamic module that precedes the ReAct reasoning loop. At every step, the module receives the triple $\{\mathcal{C}, \mathcal{P}, \mathcal{H}_t\}$ where $\mathcal{C}$ is the current conversational context, $\mathcal{P}$ the full domain-level policy set, and $\mathcal{H}_t$ the trajectory produced so far. It returns a task-focused policy subset $\mathcal{P}^f \subseteq \mathcal{P}$ and a filtered trajectory $\mathcal{H}_t^f$ by removing irrelevant regulations and hallucinated reasoning steps, thereby shrinking the search space and reducing hallucination.

As illustrated in Figure 3, whenever the agent issues an action $\alpha_i = \texttt{talk\_to\_user}$ and receives a new user utterance, the Dynamic Module is invoked before the next reasoning step. The downstream ReAct agent then proceeds with the updated input $\{\mathcal{F}, \mathcal{Q}, \mathcal{P}^f, \mathcal{T}, \mathcal{H}_t^f\}$, and this refinement is repeated after each user interaction, yielding a context-adaptive reasoning process.

## 4.2 E-PLAN&SOLVE

E-Plan-and-Solve extends the vanilla Plan-and-Solve Wang et al. (2023) paradigm as depicted in Figure 3. The original pipeline first consumes the tuple $\{\mathcal{F}, \mathcal{Q}, \mathcal{P}, \mathcal{T}\}$ to generate a high-level plan $P$. At step $t$ the vanilla Plan-and-Solve agent receives the tuple, where $\mathcal{P}$ denotes the Domain policy, and $\mathcal{H}_t$ the reasoning trajectory accumulated over the preceding $t$ steps,

$$\mathcal{H}_t = (\alpha_0, o_0, \alpha_1, o_1..., \alpha_{t-1}, o_{t-1}).$$

Here, $\alpha_i$ stands for the action at step $i$, $o_i$ represents the feedback from the action at step $i$.

To counteract policy complexity and query ambiguity, we prepend a Dynamic Module to the Plan-and-Solve pipeline. Given the triple $\{\mathcal{C}, \mathcal{P}, \mathcal{H}_t\}$ where $\mathcal{C}$ is the current conversational context, $\mathcal{P}$ the full domain policy, and $\mathcal{H}_t$ is the executed trajectory, the module emits a task-focused policy subset $\mathcal{P}^f \subseteq \mathcal{P}$ and a revised plan $P^f$. By discarding irrelevant regulations and dynamically rewriting the plan, it shrinks the search space, suppresses hallucination, and reduces the interference caused by ambiguous user intent.

As illustrated in Figure 3, whenever the agent issues $\alpha_i = \texttt{talk\_to\_user}$ and receives a new user utterance, the Dynamic Module is invoked before the next planning or reasoning step. The downstream agent then proceeds with the updated input $\{\mathcal{F}, \mathcal{Q}, \mathcal{P}^f, \mathcal{T}, P^f, \mathcal{H}_t\}$ and this refinement is repeated after every user interaction, yielding a context-adaptive planning-and-execution loop.

## 5 EXPERIMENT

### 5.1 EXPERIMENTAL SETTING

The performance of current state-of-the-art multimodal LLMs is evaluated on Mix-ECom. The evaluation employs the ReAct and Plan&Solve frameworks, utilizing GPT-4o OpenAI (2024), Gemini-2.5-Pro Google (2025), Claude-4-sonnet Anthropic (2025), and Qwen-VL-MAX Bai et al. (2023) as model backbones. Furthermore, the proposed dynamic module is integrated into these frameworks for additional assessment.

### 5.2 EVALUATION METRICS

The performance of agents on our dataset is evaluated using a method based on the comparison between Key Answers and the Database.

**Key Answer Score.** Model-generated dialogue content and the set of key answers are input, which are denoted as $\kappa = \{k_1, k_2, ..., k_n\}$. GPT-4o is employed to evaluate whether each key answer appears in the dialogue. The Key Answer Score is assigned a value of 1 if and only if all key answers in $\kappa$ are present in the dialogue. Mathematically, this can be expressed as:

$$S_{\text{ka}} = \begin{cases} 1 & \text{if } \forall k_i \in \kappa, k_i \in D \\ 0 & \text{otherwise} \end{cases}$$

where $D$ represents the dialogue text.

**Database Score.** We compare the database state obtained after the tool execution by the model with the ground truth database state. The Database Score is assigned a value of 1 if and only if the two states are entirely identical. Due to potential variations in the expression of content within the remark field (e.g., remarks specifying a logistics brand), we employ GPT-4o to determine the equivalence of this specific field.

The prompts used for the comparison in both the Key Answer Score and the Database Score can be found in Appendix B.3.

### 5.3 BASELINE MODEL

To further validate the effectiveness of training data, we conduct training on the Qwen-2.5-VL-7B Team (2025) model. In training data, an inference trajectory of step t can be represented as

$$\mathcal{H}_t = (\tau_0, \alpha_0, o_0, \tau_1, ..., \tau_{t-1}, \alpha_{t-1}, o_{t-1}).$$

| Model | Framework | Logistics | | | After-sales | | | Pre-sales | | | Total | | |
|---|---|---|---|---|---|---|---|---|---|---|---|---|---|
| | | KA. | DB. | Score | KA. | DB. | Score | KA. | DB. | Score | KA. | DB. | Score |
| GPT-4o | ReAct | 57.8 | 78.0 | 46.7 | 40.6 | 71.4 | 32.9 | 49.0 | 100.0 | 49.0 | 49.5 | 83.3 | 43.1 |
| | E-ReAct | 67.6 | 80.5 | 54.6 | 47.3 | 65.9 | 36.2 | 55.0 | 100.0 | 55.0 | 57.2 | 82.6 | 49.2 |
| | Plan&Solve | 48.1 | 76.6 | 37.0 | 38.4 | 51.6 | 24.1 | 57.0 | 100.0 | 57.0 | 48.2 | 76.9 | 39.8 |
| | E-Plan&Solve | 51.9 | 78.7 | 40.7 | 42.9 | 50.5 | 29.7 | 60.0 | 100.0 | 60.0 | 51.8 | 77.3 | 43.8 |
| Gemini-2.5-pro | ReAct | 61.2 | 82.4 | 53.7 | 56.0 | **84.6** | 48.3 | 58.0 | 100.0 | 58.0 | 58.5 | 89.0 | 53.5 |
| | E-ReAct | 75.9 | 85.8 | 67.9 | 63.7 | **84.6** | 50.5 | 62.0 | 100.0 | 62.0 | **67.6** | 90.3 | 60.5 |
| | Plan&Solve | 66.7 | 88.9 | 62.0 | 57.1 | 75.8 | 49.5 | 64.0 | 100.0 | 64.0 | 62.9 | 88.6 | 58.9 |
| | E-Plan&Solve | 71.3 | **91.7** | 66.7 | 58.2 | 79.1 | 53.8 | **65.0** | 100.0 | **65.0** | 65.2 | **90.6** | **62.2** |
| Claude-4-Sonnet | ReAct | 75.9 | 80.6 | 63.0 | 64.8 | 72.5 | 53.8 | - | - | - | - | - | - |
| | E-ReAct | **77.7** | 85.2 | **69.4** | 68.1 | 82.4 | 57.1 | - | - | - | - | - | - |
| | Plan&Solve | 70.4 | 85.2 | 62.0 | 67.0 | 79.1 | 54.9 | - | - | - | - | - | - |
| | E-Plan&Solve | 74.1 | 84.2 | 64.8 | **68.1** | 81.3 | **58.2** | - | - | - | - | - | - |
| Qwen-VL-MAX | ReAct | 58.3 | 75.4 | 42.6 | 53.8 | 64.8 | 43.9 | 56.0 | 100.0 | 56.0 | 56.2 | 80.3 | 47.5 |
| | E-ReAct | 55.6 | 77.8 | 44.4 | 52.7 | 71.4 | 46.1 | 57.0 | 100.0 | 57.0 | 55.2 | 83.3 | 49.2 |
| | Plan&Solve | 48.1 | 78.7 | 36.1 | 42.8 | 51.6 | 26.4 | 51.0 | 100.0 | 51.0 | 47.5 | 77.6 | 38.1 |
| | E-Plan&Solve | 51.9 | 80.6 | 38.9 | 46.2 | 50.5 | 28.6 | 58.0 | 100.0 | 58.0 | 52.2 | 77.9 | 42.1 |
| Qwen-2.5-VL 7B | ReAct | 2.7 | 45.4 | 0.9 | 2.2 | 29.6 | 0.0 | - | - | - | - | - | - |
| Qwen-2.5-VL 7B* | ReAct | 29.4 | 62.0 | 19.3 | 24.3 | 49.4 | 17.7 | - | - | - | - | - | - |

Table 3: Evaluation results for MLLMs. "KA." denotes the Key-Answer score, i.e., the fraction of tasks whose key answers are all correct. "DB." denotes the Database score, i.e., the fraction of tasks whose database results are correct. "Score" is the fraction of tasks on which both the key answers and the database are correct. "*" means that the model is fine-tuned. "-" indicates video modality is not supported. The results are presented in percentage (%).

Here, $\tau_i$ denotes the thinking process at step $i$, $\alpha_i$ stands for the action taken at step $i$, $o_i$ represents the feedback resulting from the action at step $i$.

We split each $\mathcal{H}_t$ into $t$ segments. For each segment, we construct supervised fine-tuning data by using $\mathcal{H}_{i-1} = (\tau_0, \alpha_0, o_0, \tau_1, ..., \tau_{i-1}, \alpha_{i-1}, o_{i-1})$ as the instruction and $(\tau_i, \alpha_i)$ as the output. Due to resource constraints, we only train the model on the Logistics and After-sales subsets, yielding the "Qwen-2.5-VL Trained" model.

Due to the inclusion of video content in pre-sales tasks and owing to resource constraints, we only train Qwen-2.5-VL on data related to logistics and after-sales tasks.

## 5.4 MAIN RESULT

Table 3 shows that Gemini 2.5 pro achieves the highest overall score (62.2) under the E-Plan-and-Solve framework, followed closely by Gemini-2.5-Pro with 60.5 under E-ReAct. Qwen-VL-Max and GPT-4o lag substantially behind. Nevertheless, even the best-performing model remains far from solving our benchmark.

Across GPT-4o and Qwen-VL-Max, ReAct outperforms Plan-and-Solve, whereas for Gemini-2.5-Pro and Claude-4-Sonnet the relation reverses or becomes comparable. Consistently, E-ReAct improves upon ReAct, and E-Plan-and-Solve improves upon Plan-and-Solve for every model, corroborating the effectiveness of our enhancements. The largest gains appear in Logistics tasks, where user queries are concise and the rule-filtering module seldom hallucinates; improvements on pre-sales and after-sales tasks are more modest.

Owing to task complexity and input lengths that exceed the pre-training context of Qwen-2.5-VL-7B (domain policies, tool descriptions, and multi-modal files), the base model scores only 0.9 on Logistics and 0 on after-sales. After supervised fine-tuning on our data, its scores rose to 19.3 and 17.7, respectively, verifying the utility of the curated dataset.

Table 4 presents the results of the human evaluation, which was conducted by five professional e-commerce customer service staff across three key dimensions: Human-likeness, Informativeness, and Key Answer. The experimental results indicate that Gemini-2.5-pro attained the highest scores across all three dimensions.

|  | GPT-4o | Gemini-2.5-pro | Claude-4-sonnet | Qwen-VL-MAX | Ground Truth |
|---|---|---|---|---|---|
| Human-likeness | 60.4 | **69.2** | 67.8 | 49.2 | 82.6 |
| Informativeness | 66.8 | **74.6** | 70.2 | 62.0 | 91.8 |
| Key Answer | 59.4 | **68.4** | 66.2 | 54.6 | 100.0 |

Table 4: Human evaluation results for various MLLMs.

| Multi-modal | Rule | Logistcs | After-sales | pre-sales | Model | Logistics | After-sales |
|---|---|---|---|---|---|---|---|
| ✓ | ✓ | 46.7 | 32.9 | 49.0 | GPT-4o | 60.9 | 55.6 |
| ✗ | ✓ | 46.7 | 28.9 | 43.0 | Gemini-2.5-pro | 68.4 | 42.1 |
| ✓ | ✗ | 2.2 | 17.6 | 37.0 | Claude-4-sonnet | 54.2 | 37.5 |

Table 5: Results of GPT-4o on different settings, with or without multi-modal input and rules.

Table 6: The percentage of failure cases that do not contain all key answers.

## 5.5 BASE CASE ANALYSIS

The failure modes of a GPT-4o-backed ReAct agent within the benchmark are analyzed and categorized into four types:

**Multimodal Misinterpretation.** Approximately 15 % of GPT-4o failures stem from misunderstanding multimodal evidence. In Appendix D.1, for example, the user uploads a photo that clearly justifies an after-sales request; the model nevertheless claims the image is inconclusive and denies the claim. In the during-sales subset, the agent is unable to extract product information from live-stream clips containing regional dialect, and thus fails to answer the customer's question. To quantify the impact of multimodality, we stripped all non-text inputs and re-evaluated GPT-4o. As Table 5 shows, its score drops by only 3.3 and 6.0 on the two splits, indicating that the model barely exploits visual cues. This also shows that our dataset presents a tough challenge for function-calling agents: there is still a lot of work left to do when it comes to understanding complex multi-modal content and making correct decisions.

**Violation of Domain Rules.** Roughly 63 % of errors arise from disregarding fine-grained policies. Appendix D.2 gives a logistics instance: when an in-transit order is subject to address change, the Rules requires updating both the order-level shipping address and the courier-level destination while resetting the logistic status. To further verify the impact of Domain Rules on the model, we conducted tests on the Logistics and After-sales subsets with the Domain Rules removed. The results are shown in Table 5, the scores decreased by 44.5 and 11.3, respectively. This demonstrates the importance of Domain Rules, and how to enable agents to comply with complex, fine-grained rules remains a key challenge in current research.

**Premature Switch to Human.** About 12 % of failures are due to unnecessary human hand-off. In one logistics task, the shipping-fee calculator fails because the delivery address is missing; instead of querying the user via `talk_to_user`, the agent gives up and escalates. In an after-sales dialogue, after the customer refuses a coupon compensation, the correct next action is to propose an alternative remedy (e.g., return & refund), yet the agent again transfers to a human. These cases highlight the subtlety of judgment of last resort in customer service: knowing when to escalate is as critical as knowing how to solve the problem.

**Other Errors** The remaining 5 % include malformed tool calls that trigger infinite loops, user agent to articulate explicit demands, and partial key-answer generation. Table 6 reports the proportion of failures in which the agent nevertheless hits some required key answers.

## 6 CONCLUSION

This work introduced Mix-ECom-Bench, a benchmark designed to evaluate the capabilities of LLM agents. The benchmark integrated four dialogue types (QA, recommendation, task-oriented dialogue, and chitchat), three e-commerce task categories (pre-sales, logistics, and after-sales), and 82 e-commerce rules along with 4,799 real-world e-commerce dialogues. Furthermore, baseline models

were evaluated on Mix-ECom-Bench, and this work proposed a dynamic e-commerce framework to address the associated challenges. Results show that current e-commerce agents lack sufficient capabilities to handle e-commerce dialogues, due to the hallucination cased by complex domain rules.

## ETHICAL STATEMENT

We make sure that Mix-ECom is collected in a manner that is consistent with the terms of use of any sources and the intellectual property and privacy rights of the original authors of the texts. And the crowd workers were treated fairly. This includes, but is not limited to, compensating them fairly, ensuring that they were able to give informed consent, and ensuring that they were voluntary participants who were aware of any risks of harm associated with their participation.

## ACKNOWLEDGEMENTS

This work was supported by the National Science Fund for Distinguished Young Scholars (No. 62025603), the Fundamental Research Funds for the Central Universities, the National Natural Science Foundation of China (No. 62576299, No. U21B2037, No. U22B2051, No. U23A20383, No. 62176222, No. 62176223, No. 62176226, No. 62072386, No. 62072387, No. 62072389, No. 62002305, No. 62272401), and the Natural Science Foundation of Fujian Province of China (No. 2021J06003, No. 2022J06001).

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

APPENDIX

# A   THE USE OF LARGE LANGUAGE MODELS

In this work, LLMs are employed to polish and enhance the writing style of the paper.

# B   PROMPT

## B.1   DOMAIN RULES

```
# Basic Guidelines

- Please respond to customers' needs politely, patiently, and
    professionally, using the tone and vocabulary of an e-commerce
    customer service representative.

- The current system time is 00:00 on June 12, 2025.

- If you encounter problems that cannot be resolved using the available
    tools, promptly use the switch_to_human tool to transfer to a human
    agent, rather than giving users meaningless responses.

- If a user shows strong negative emotions, use the switch_to_human tool
    to transfer to a human agent.

- Do not ask users for any ID information; any ID information you need
    will be provided in subsequent queries.

- If the provided ID information does not include an order_id, it means
    the user has not yet placed an order. If there is no logistics_id, it
    means the seller has not yet shipped the order.

- For special customer requests, such as expedited shipping, specified
    delivery times, or other product-related special requirements, try to
    record them in the notes.

- After resolving the user's current request, please confirm if there are
    any other needs.

- If the user indicates that all requests have been resolved and you have
    already used the tools to complete all user requests, please call
    the end_conversation tool to end the current conversation.

# Shipping Cost Calculation Guidelines

- When calculating total shipping costs, consider the total weight of the
    goods, i.e., quantity * unit weight.

- When calculating return shipping costs, use the same logistics brand
    used for the original shipment.

- If there is no current logistics information (order not shipped),
    compare the logistics brands used by the merchant for return shipping
    costs and calculate using the cost of the cheapest brand.

- When users ask if they need to advance shipping costs, inform them of
    the specific amount they need to advance, considering the shipping
    insurance situation.

- If the return is due to the merchant's fault, the user does not need to
    pay return shipping costs, regardless of whether shipping insurance
    is included.
```

- If a logistics interception is due to an address change, the user does not bear additional shipping costs.

# Shipping Insurance Guidelines

- Shipping insurance is a service offered by some merchants to subsidize return shipping costs during returns.

- The maximum subsidy amount for shipping insurance is 9 RMB. If the return shipping cost exceeds this amount, the user must cover the difference.

- When users inquire about advancing shipping costs, calculate the shipping cost first, then answer based on the shipping insurance situation.

# Package Signed For But Not Received

- If a user reports that an order is marked as delivered but not received, advise them to check with family/friends or contact the logistics company.

# Logistics Brand Selection Guidelines
- Merchant acceptance of specified brands means choosing from among the several brands they use, not specifying any arbitrary brand.

- For already shipped orders, specifying or changing the logistics brand is not allowed.

- If the merchant supports specifying a logistics brand and the user specifies a brand used by the merchant, record this requirement in the notes.

- If the merchant does not support specifying logistics brands, or the customer's specified brand is not among those used by the merchant, politely decline the request.

# Logistics Time Calculation Guidelines

- The current system time is 00:00, Thursday, June 12, 2025.

- When replying about time, use the format "Month Day Hour", e.g., "Your estimated arrival time is 13:00 on June 12". All time calculations should be precise to the hour.

# Estimated Shipping Time Calculation Guidelines

- For placed orders (with order_id), the estimated shipping time is the order payment time plus the merchant's promised shipping time.

- For orders not yet placed (no order_id), the estimated shipping time is the current system time plus the merchant's promised shipping time.

# Estimated Arrival Time Calculation Guidelines
- For shipped orders (with logistics_id), the estimated arrival time is the "logistics pickup time" plus the "logistics transit time".

- For orders not yet shipped (no logistics_id), the estimated arrival time is the "estimated shipping time" plus the "logistics transit time".

- For orders not yet shipped (no logistics_id), if the merchant does not support specifying a logistics brand, or if the user did not

successfully specify one, compare the transit times of the logistics
   brands used by the merchant and use the longest time for estimation.

- If the user successfully specified a logistics brand, use that brand's
   transit time for estimation.

# Shipping Address Modification Guidelines

- If the order has not been shipped (no logistics_id), the shipping
   address in the order database (receive_address) can be directly
   modified to the new address.

- If the order has been shipped, modifying the shipping address requires
   confirming the current logistics status, which can be [In Transit,
   Delivered].

- If the logistics status is "In Transit", initiate a logistics
   interception. This involves three actions: 1. Modify the
   receive_address in the order database to the new address. 2.Modify
   the receive_address in the logistics database to the new address. 3.
   Change the logistics status in the logistics database to "Intercepted
   ".

- If the logistics status is "Delivered", negotiate a return process with
    the user; see Return Guidelines for details.

- If the logistics status is "Delivered" and the user's situation does
   not meet the return conditions, inform the user to contact the
   logistics company.

# After-Sales Guidelines

- When users raise after-sales requests for the following reasons:
   missing items/wrong items received, goods damaged during transit,
   dissatisfaction with product quality, first guide the customer to
   send relevant pictures to verify their claim.

- Content in the User's message formatted like [Image x] represents the
   user sending an image at that point, corresponding to the x-th image
   in the input. For example, [Image 2] means the second image provided
   to you.

- If the user wants to return an item for personal reasons (e.g., don't
   like it, don't want it, bought too many, wrong item), do not require
   them to send pictures.

- If the user cannot provide proof pictures, or the provided pictures
   cannot verify the after-sales claim, first comfort the user and
   politely decline the request.

- If the reason for the after-sales request is missing items, after
   verifying with pictures, inform the customer that the missing items
   will be resent and record this in the notes.

- If the reason is damage during transit or dissatisfaction with product
   quality, after verifying with pictures, first attempt to resolve the
   issue by offering a small red envelope compensation to reach a
   settlement.

- The maximum red envelope compensation amount is Order Payment Amount *
   Merchant's Maximum Compensation Percentage, rounded down to the
   nearest whole number, with a minimum compensation of 1 RMB.

- Do not inform the user of the specific calculation method for the red
   envelope compensation.

- If the user accepts the red envelope compensation, record the
  compensation details in the notes.

- If the red envelope compensation cannot resolve the user's issue,
  negotiate entering the return process; see Return Guidelines.

- If neither red envelope compensation nor return can resolve the user's
  issue, transfer to a human agent.

## Reshipment Guidelines
- Reshipment can only be registered for the customer if it is verified
  that the merchant indeed shipped missing/wrong items. Record the
  reshipment details in the notes.

- Except for cases of missing/wrong items shipped by the merchant,
  reshipment cannot be registered under any other circumstances.

- If the customer requests reshipment but does not meet the conditions,
  politely inform them of this result.

## Red Envelope Compensation Guidelines
- The maximum red envelope compensation amount is Order Payment Amount *
  Merchant's Maximum Compensation Percentage, rounded down to the
  nearest whole number, with a minimum compensation of 1 RMB.

- Do not inform the user of the specific calculation method for the red
  envelope compensation.

- If the user accepts the red envelope compensation, record the
  compensation details in the notes.

## Return Guidelines

- To initiate a return process, first determine if the product belongs to
  the fresh/perishable goods category. If it is, follow the procedures
  under the Refund-Only Guidelines.

### Personal Reason Return Guidelines

- If the user's request is for a return due to personal reasons, you do
  not need to guide them to provide photo proof.

- Since the return is for personal reasons, the maximum service that can
  be provided is a return & refund, if the conditions are met (see
  Return & Refund Guidelines).

- Do not process refund-only, reshipment, or red envelope compensation
  for personal reason returns.

### Product Quality Reason Return Guidelines

- If the user's request is a complaint and return due to product quality
  issues, first guide the customer to send pictures to verify the
  reason for the complaint.

- If the user's pictures can verify the complaint, first attempt to use a
  small red envelope compensation to comfort the customer. If
  compensation cannot resolve the issue, negotiate entering the return
  & refund process (see Return & Refund Guidelines). In this case, as
  it is the merchant's fault, inform the user that they do not need to
  pay return shipping costs.

- If the user's pictures cannot verify the complaint, politely comfort
  the user and, based on the specific content of the pictures, politely

decline requests for red envelope compensation, refund-only, or
reshipment. If the customer's demand is a return & refund, proceed to
the return & refund process, noting that this should be treated as a
personal reason return, and inform the user of the return shipping
cost based on shipping insurance.

### Return & Refund Guidelines

- If the user has already returned the goods, red envelope compensation
cannot be provided.

#### If the order is not yet shipped (i.e., no logistics_id)

- Directly change the order status in the order database to "Cancelled".

#### If the order is already shipped

- Before processing the return, confirm whether the product supports 7-
day no-reason returns (is_support_7d_back) and confirm with the user
whether the product has been used.

- A return can only be processed if the product has not been used, and
the product supports 7-day no-reason returns, and the time since
receipt is less than or equal to 7 days.

- If the user meets the return conditions, you need to provide the user
with the merchant's address so they can initiate the return shipment.
You also need to inform the customer about shipping insurance
information and the specific amount they need to advance for shipping
(accurate to one decimal place).

- You can only process the return after the user informs you that they
have initiated the return shipment.

##### User Level is 3

- Process an expedited return for the user: change the order status in
the order database to "Refunded".

###### User Level is below 3

-Process a standard return for the user: change the order status in the
order database to "Returning".

### Refund-Only Guidelines

- Note: Refund-only can only be processed for fresh/perishable category
goods. Other product types can only undergo the return & refund
process.

- For fresh/perishable goods, product issues must be verified before
processing a refund-only. If a refund-only is initiated, change the
order status to "Refund-Only".

- For fresh/perishable goods, confirming whether the product has been
used is not required for refund-only processing.

- For verified issues with fresh/perishable goods, do not process a
return & refund for the customer; process a refund-only.

- When processing a refund-only, it is not necessary to provide the user
with the merchant's address, nor does the customer need to initiate a
return shipment.

```
- If the customer requests a refund-only but does not meet the conditions
    , politely inform them of this result.

# Coupon Guidelines
- A coupon can be used for a current product only if the following
    conditions are met: 1. The current product type is included in the
    coupon's category_list. 2. The current product's price meets the
    coupon's minimum_purchase condition.

- Coupons of different levels can be used together. Coupons of the same
    level cannot be stacked.

- If there are coupons applicable to the current product, inform the
    customer of the minimum payable amount after applying the coupons.

# Product Recommendation Guidelines
- If no current products meet the user's requirements, inform the user of
     this result and recommend products that are similar to their request
     .

# Get Live Stream Clip Guidelines
- When users ask questions that you cannot determine the answer to, you
    can try using the get_item_video tool to get information from the
    product's recent live stream clips.
```

Listing 1: Domain Rules used in Mix-Ecom Benchmark

## B.2 QUESTION TYPE GENERATION FOR AFTER SALES TASKS

You will see a conversation from an e-commerce customer service scenario, where "User: xxx" represents messages sent by a customer to the e-commerce Assistant, and "Assistant: xxx" represents the Assistant's replies.
Content in the form of '[Image x]' indicates that there is an image at that point, corresponding to the x-th image I sent you. For example, "[Image 2]" represents the second image I sent you.
[Conversation Start]
{conversation}
[Conversation End]
Below is the relevant item information for this conversation.
[item Information Start]
{item_info}
[item Information End]
# Task Overview
The conversation content may relate to after-sales issues concerning a product. Your task is to extract the following information based on this: 1. The specific reason the user initiated the after-sales request. 2. Whether the images provided by the user can substantiate their claim. 3. The solution desired by the user. 4. The user's current mood.
Please strictly adhere to the following principles:
- Return the result **only** in JSON format. Return *only* this JSON result, without adding any other content. Ensure the content you return can be parsed by Python's 'json.loads()' function.
{
'reason': 'The specific reason the user initiated the after-sales request',
'image_verification': 'Whether the images provided by the user can substantiate their claim',
'solution': 'The solution desired by the user',
'mood': 'The user's current mood'
}
## The specific reason the user initiated the after-sales request

- Please select the specific reason the user initiated the after-sales request from the following options: [Missing/Wrong items shipped, Item damaged during transit, Dissatisfied with product quality, Return due to other personal reasons].
- If the user's reason does not fall into any of the above categories, write '-1'.
## Whether the images provided by the user can substantiate their claim
- If the image(s) provided by the user can substantiate their specific reason for the after-sales request (e.g., the image proves missing items, proves damage during transit, or proves the point of dissatisfaction with the product), write '1'. Otherwise, write '0'.
- You can also use the specific dialogue content to assist your judgment. For example, if the assistant does not raise objections to the user's image(s), it implies the images are valid; write '1'.
- If the conversation is incomplete, you can only judge based on the content of the image(s).
## The solution desired by the user
- Please select the solution desired by the user from the following options.
- If the desired solution cannot be determined from the conversation, or if the user's desired solution is not among the options below, please randomly select one from [Re-ship missing items, Compensate with a red envelope of x yuan, Refund and return, Refund only] to fill in.
- If the user's desired solution is compensation of x yuan, please choose an appropriate compensation amount based on the conversation or product information.
## The user's current mood
- Please infer the user's current mood based on the conversation content. Select from the following two options: [Calm, Impatient].
- If the user's current mood cannot be inferred from the conversation content, please fill in '-1'.
- Return the result **only** in JSON format. Return *only* this JSON result, without adding any other content. Ensure the content you return can be parsed by Python's 'json.loads()' function.
Now I will provide you with 4 examples:
Example 1:
{
'reason': 'Missing/Wrong items shipped',
'image_verification': '1',
'solution': 'Refund and return',
'mood': 'Impatient'
}
Example 2: {
'reason': 'Dissatisfied with product quality',
'image_verification': '0',
'solution': 'Compensate with a red envelope of 15 yuan',
'mood': '-1'
}
Example 3: { 'reason': '-1',
'image_verification': '1',
'solution': 'Refund only',
'mood': 'Calm'
}
Now, please extract based on the conversation content: 1. The specific reason the user initiated the after-sales request. 2. Whether the images provided by the user can substantiate their claim. 3. The solution desired by the user. 4. The user's current mood.
json:

## B.3 EVALUATION

### B.3.1 PROMPT FOR KEY ANSWER SCORE

You will see a message from a conversation between an e-commerce customer service representative and a customer. Your task is to determine whether the message contains the content of the key_answer.
[Start of message]
{message}
[End of message]
[Start of key_answer]
{key_answer}
[End of key_answer]
Please determine whether the message contains the content of the key_answer. If it does, reply with 1; otherwise, reply with 0.
Note: If the key_answer mentions an amount or time, the related content in the message must be an exact match to be considered correct. Only respond with 1 or 0, without providing your reasoning.
Your answer:

### B.3.2 PROMPT FOR DATABASE SCORE

You will see two messages, message1 and message2. Your task is to determine whether the content of these two messages is similar.
[Start of message1]
{remark_message}
[End of message1]
[Start of message2]
{ground_truth}
[End of message2]
Please judge whether the content of these two messages is similar. If yes, reply with 1; otherwise, reply with 0.
Note: If the messages mention express delivery brands or similar content, they will only be considered correct if the related information is completely consistent in both messages.
Reply only with 1 or 0, without providing any reasoning.
Your answer:

## C API TOOLS

| Tool Category | Included Tool Names | | Count |
|---|---|---|---|
| Information Retrieval | get_shop_detail
get_item_detail
get_user_detail
get_product_detail | get_order_detail
get_logistics_detail
get_user_coupon_detail
get_video_detail | 8 |
| Information Calculation | calculate_shipping_time | calculate_shipping_cost | 2 |
| Information Modification | modify_logistics_address
modify_order_address | modify_logistics_state
modify_order_state | 4 |
| Interaction & Management | talk_to_user
remark | switch_to_human
end_conversation | 4 |

Table 7: Tools in MIX-ECOM Bench.

# D   SHOW CASE

Please see the next page.

## D.1 Case1

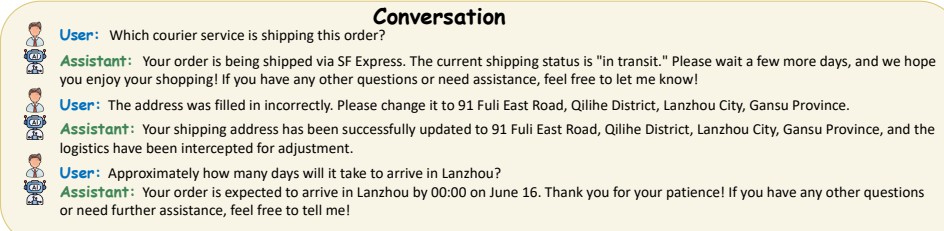

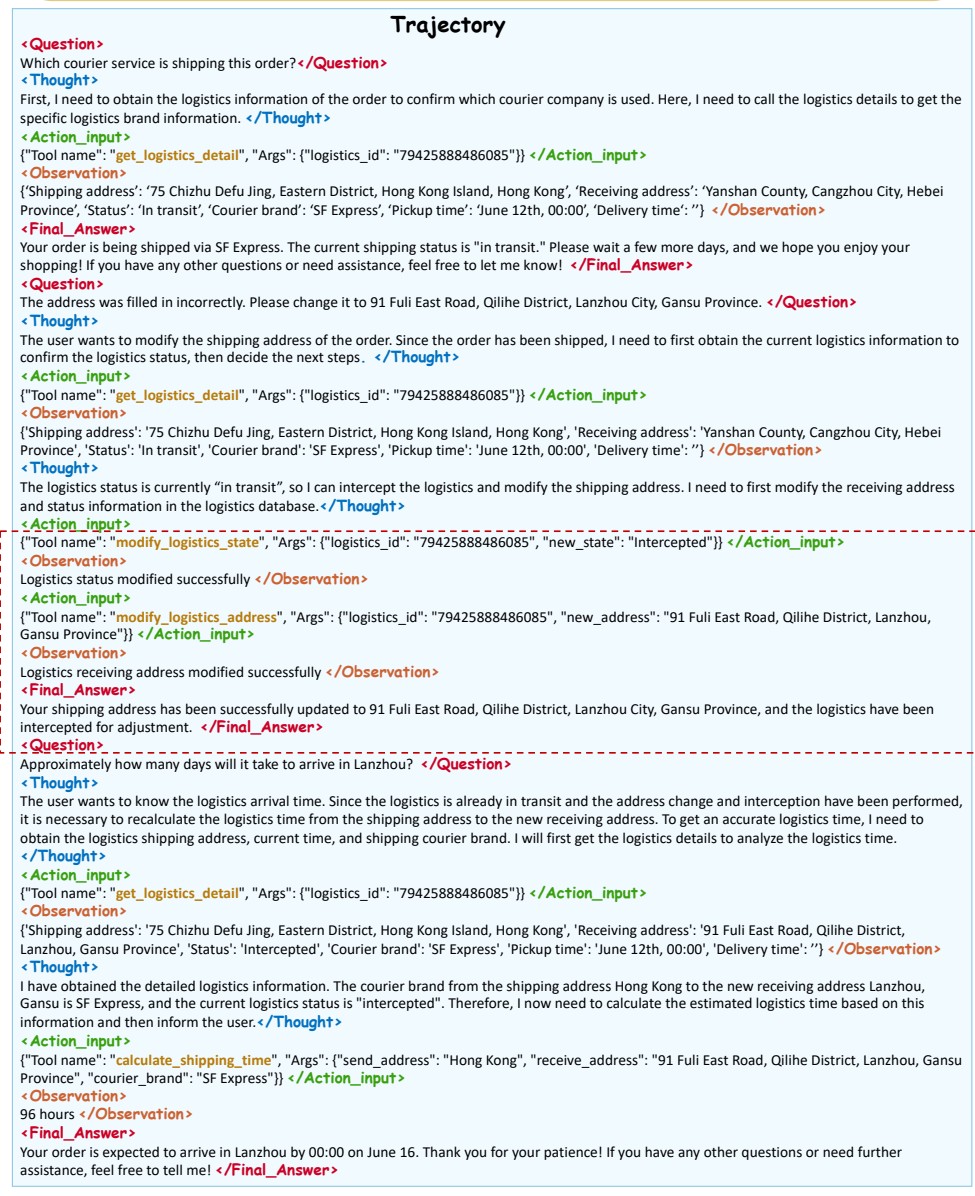

Figure 4: The assistant does not follow the domain rule regarding the modification of the receiving address when the logistics status is 'in transit'. According to the rule, the assistant must modify three pieces of information: the logistics status, the logistics address, and the order address. However, the assistant only modifies the logistics status and the logistics address, which leads to task failure.

## D.2 CASE2

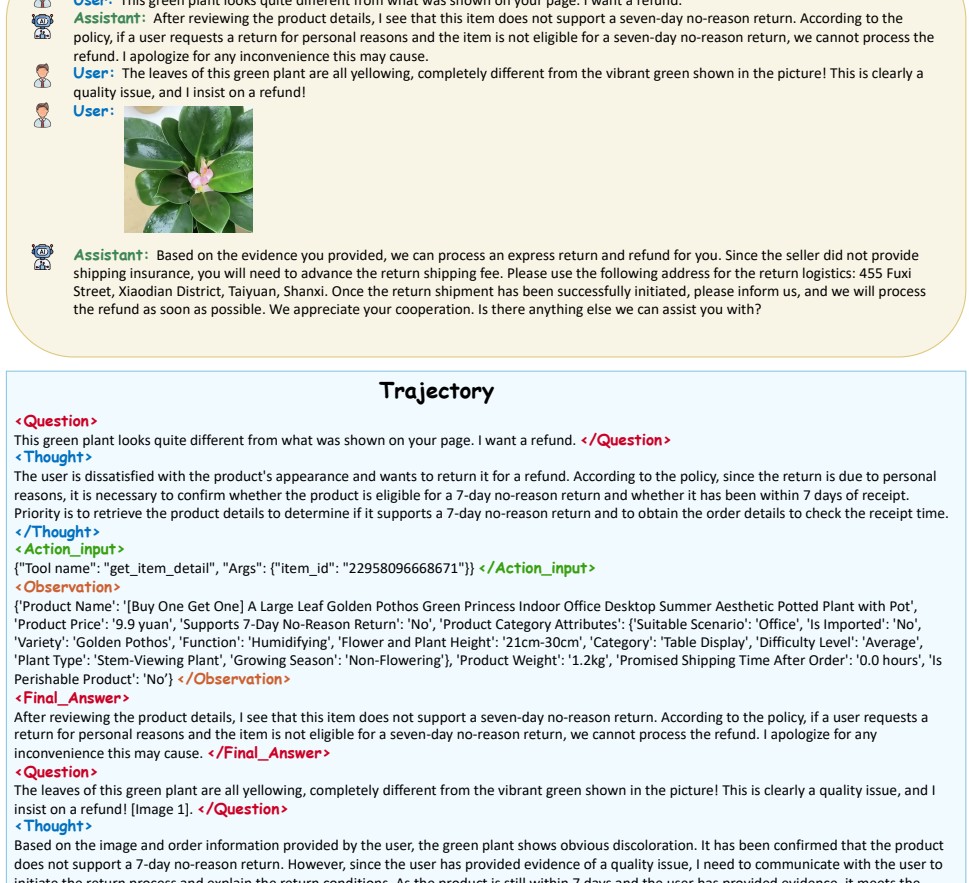

Figure 5: The assistant does not utilize the information in the image to make the correct decision. In this case, the unreasonable request of the user should be rejected; however, the assistant processes the refund due to the repeated demands of the user, resulting in task failure.

