# OpenReview forum: "Mix-Ecom: Towards Mixed-Type E-Commerce Dialogues with Complex Domain Rules"
_ICLR.cc/2026/Conference — ICLR 2026 Poster_

### Official Review · Reviewer_bcjS · 2025-10-29

**Soundness:** 3
**Presentation:** 3
**Contribution:** 2
**Rating:** 2
**Confidence:** 3

**Summary:**

This paper introduces Mix-ECom, a novel dataset and benchmark aimed at evaluating the performance of e-commerce agents in handling real-world, mixed-type e-commerce dialogues. These dialogues involve various types, including task-oriented, recommendation, Q&A, and chit-chat, while adhering to complex domain rules in pre-sales, logistics, and after-sales tasks. The dataset includes 4,799 dialogue samples, covering 82 e-commerce rules, and integrates multi-modal data (e.g., images and videos) to evaluate agents' capabilities. The paper proposes a dynamic framework for improving e-commerce agent performance, addressing the problem of hallucinations arising from complex rules.

**Strengths:**

The Mix-ECom dataset is designed to address the lack of comprehensive benchmarks for mixed-type e-commerce dialogues with complex rules. The inclusion of multi-modal content is a valuable feature that enhances the practical applicability of the dataset.
The focus on real-world e-commerce dialogues, such as pre-sales, logistics, and after-sales, ensures that the research is directly relevant to industry applications. This has strong potential for real-world impact, especially in enhancing customer service experiences through intelligent agents. The experiment compares some recent e-commerce framework, using the Mix-ECom versus not using it, the exp demonstrated the improvement brought by the Mix-ECom.

**Weaknesses:**

1.The paper does not provide a clear presentation of the dataset's distribution across different task categories, dialogue types, or domain rules. This information is crucial for understanding the dataset’s structure and its potential biases. The absence of such details makes it difficult to evaluate the dataset's representativeness and fairness.

2.While the dataset and framework are innovative, the paper does not provide sufficient early-stage details on the dataset construction process, usage information, or any plans for ongoing updates. Without such transparency, it is unclear how this work will benefit the community in the near future or whether other researchers can effectively build upon this resource.

3.The paper lacks clarity regarding the data collection process, particularly the sources of the dialogues and whether proper permissions and licenses were obtained. The absence of such details raises concerns about data privacy and legal compliance, which could impact the credibility and applicability of the dataset in real-world scenarios.

**Questions:**

refers to weakness paragraph

---

> ### Author Response · Authors · 2025-11-17
>
> # W1: The paper does not provide a clear presentation of the dataset's distribution across different task categories, dialogue types, or domain rules.
>
>
> 1. In Section 3.4 and Table 2 of the paper, we have provided a detailed introduction to DATA STATISTICS, which includes the distribution of task categories.
> 2. In Appendix B.1, we have presented the complete Domain Rules, which are applicable to all task scenarios. These Domain Rules are drafted based on the basic e-commerce business logic of the partner e-commerce company. The following table shows the distribution of domain rules related to tasks in our evaluation set.
> | Guidelines name                          | Basic | Shipping Cost Calculation | Shipping Insurance | Package Signed For But Not Received | Logistics Brand Selection | Logistics Time Calculation |
> |------------------------------------------|-------|----------------------------|--------------------|--------------------------------------|---------------------------|----------------------------|
> | percentage                               | 100.0%| 18.1%                      | 31.4%              | 6.4%                                 | 24.7%                     | 13.0%                      |
>
> | Guidelines name                          | Shipping Address Modification | Reshipment Guidelines | Compensation Guidelines | Return Guidelines | Personal Reason Return | Product Quality Reason Return | Coupon Guidelines | Product Recommendation |
> |------------------------------------------|-------------------------------|-----------------------|-------------------------|-------------------|-------------------------|-------------------------------|-------------------|-------------------------|
> | percentage                               | 19.1%                         | 6.4%                  | 8.7%                    | 19.1%             | 8.7%                    | 18.4%                         | 6.7%              | 10%                     |
>
> 3. Mixed-type refers to the fact that our tasks consist of the following four types: Task-oriented Dialogue, Recommendation, Question & Answer, and Chitchat. These four types of tasks have been clearly defined in traditional NLP dialogue systems. The following table shows the distribution of task types in our evaluation set.
>
> | Task type          | Task-oriented Dialogue | Recommendation | Question & Answer | Chitchat |
> |--------------------|------------------------|---------------|-------------------|----------|
> | percentage         | 78.3%                  | 33.4%         | 79.9%             | 64.9%    |
>
> # W2: While the dataset and framework are innovative, the paper does not provide sufficient early-stage details on the dataset construction process, usage information, or any plans for ongoing updates.
>
> In Section 3.4 and Figure 2 of this paper, we have provided a detailed explanation of the data construction process.
>
> We speculate that the "early-stage details" you mentioned might refer to the original e-commerce customer service dialogue content and user profile information. The original dialogue content source is the real business scenarios of the partner e-commerce company, while the user profile information is derived from the company's private data.
>
> If you could further clarify what specifically the "early-stage details" refer to, we will promptly supplement the relevant information in our subsequent responses.
>
> Regarding the use and release plan for the data, since it involves the review process of the partner company's legal and security departments, we plan to open-source the data after the rebuttal phase ends and the data is finalized. We will continue to iterate and release new versions to improve the data quality and scale, and explore the effectiveness of the training data under the SFT + reinforcement learning training paradigm.
>
> # W3: The paper lacks clarity regarding the data collection process, particularly the sources of the dialogues and whether proper permissions and licenses were obtained.
>
> Thank you for your valuable suggestions. This project was completed in collaboration with a well-known e-commerce company, and all the data used are derived from the company's real business data.
> Regarding privacy processing, as shown in steps five and six in Figure 2, LLM rewriting and manual screening are applied:
> 1. For sensitive information such as user and merchant names, addresses, and phone numbers that may appear in the conversation, Qwen is utilized to uniformly delete or replace these fields to ensure privacy security.
> 2. In the manual screening phase, manual processing is applied to further filter and remove data that the LLM fails to successfully rewrite for privacy.
>
> In addition, all the data  pass the joint review of the security and legal departments of the partner company and will be publicly available for further research.

---

### Official Review · Reviewer_DZnq · 2025-10-30

**Soundness:** 2
**Presentation:** 3
**Contribution:** 2
**Rating:** 4
**Confidence:** 3

**Summary:**

This work mainly tries to fill the gap of handling mixed-type dialogues and complex domain rules in the context of LLM-based e-commerce agent benchmarks.

To this end, this work has introduced a new dataset Mix-ECom, which is derived from 70,000 real-world customer-service dialogues and finally keeps about 4799 samples.  The highlights of Mix-ECom can be summarized as:

1. It includes four dialogue types (QA, recommendation, task-oriented dialogue, chit-chat).
2. It includes three e-commerce task categories (pre-sales, logistics, after-sales).
3. It includes 82 domain rules.

Then, this work has proposed a framework E-ReAct by tailoring the ReAct work and an E-Plan-and-Slove module extended from Plan-and-Solve.

In the experiments, the authors evaluate 5 LLMs (4 closed-source, 1 open-source) on Mix-ECom.

**Strengths:**

1. This work has introduced  an novel dataset Mix-ECom,  addressing the gap of mixed-type dialogues and complex rules in existing e-commerce agent evaluations.  The dataset construction has used real user dialogues and proper construction pipelines.

2. Compared to the previous benchmark dataset, Mix-ECom is able to cover more practical scenarios.

3. Mix-ECom has covered  multi-modal inputs (images, videos).

4. Conducts comprehensive experiments with 5 LLMs (4 closed-source, 1 open-source) and fine-tuning, providing clear results and highlighting current agents’ limitations.

5. Compared to the original ReAct and Plan&Solve, the proposed E-Version is much better in the experiments.

**Weaknesses:**

1. The major contribution of this work is the proposed dataset. However, the novelty and contribution of this dataset are not very significant.
2. The dataset's development and statistical analysis follow standard practices, lacking notable innovation or depth.
3. E-ReAct and E-Plan&Solve are incremental because both of them are derived from other works without notable modification.
4. Only 0-shot experiments are conducted and only 5 LLMs are tested. It is not sufficient enough for a dataset paper.
5. No enough analysis of how fine-tuning (e.g., on Mix-ECom subsets) impacts performance, which may limit understanding of the benchmark’s value.
6. No comparison with specialized e-commerce dialogue models.
6. No manual evaluation in the experiment part.

**Questions:**

1. Have you considered testing against specialized e-commerce models?

2. Do you intend to add a user study (e.g., measuring customer satisfaction with hallucination-reduced outputs) .

---

> ### Author Response · Authors · 2025-11-17
>
> # W1, W2&W3: About novelty and contribution of our work
>
> 1. This work first proposes a “mixed-type” task framework in the e-commerce customer service field. Unlike previous studies that usually focus only on specific tasks (such as recommendation or question answering), we have noticed that tasks in real e-commerce customer service scenarios are highly mixed. Therefore, the dataset we have built is closer to practical applications and has greater practical significance.
> 2. This work is completed in collaboration with a well-known e-commerce company, which serves hundreds of millions of users. The dataset construction is obtained based on its real business conversations and standard format database. Compared with similar studies, this dataset shows significant advantages in authenticity and business fit.
> 3. The data annotation work is completed by a professional team from the partner company. All annotators have extensive practical experience in e-commerce customer service, ensuring the professionalism and reliability of the annotation process and results, and also making the subsequent evaluation work more credible.
> 4. The design of the Dynamic Module is inspired by observations of the dynamic characteristics of user behavior in e-commerce conversations: user needs continue to evolve during the conversation. Therefore, the system needs to have the ability to dynamically select relevant domain rules and flexibly adjust the planning path. Experimental results show that this framework brings significant performance improvements. In fact, the E-ReAct and E-Plan-and-solve frameworks can be widely applied to various conversational systems where user intentions change frequently.
> 5. As reviewers um8L and j6Yo pointed out, this research work is of high quality, and the dataset built to simulate real conversational scenarios has positive contribution value to the academic community.
>
> # W4: Only 0-shot experiments are conducted and only 5 LLMs are tested. It is not sufficient enough for a dataset paper.
> In previous experiments using GPT4o as the backbone and ReAct as the framework, we attempted to use few-shot learning, with the results shown in the table below:
> |        | Logistics | After-sales | Pre-sales | Total |
> |--------|-----------|-------------|-----------|-------|
> | 0-shot | 46.7      | 32.9        | 49.0      | 43.1  |
> | 2-shot | 43.5      | 34.0        | 47.0      | 41.8  |
>
> According to the experimental results, compared with the 0-shot setting, the 2-shot setting did not bring significant performance improvements and even led to performance degradation in some tasks. During the evaluation process, the model needs to conduct multi-turn reasoning, with each turn including the complete domain rules and all previous reasoning trajectories, resulting in a very large token overhead. Using few-shot learning significantly increases the testing cost. Considering cost control, we did not continue with further few-shot related experiments.
>
> Given the characteristics of our tasks, such as long context lengths and extremely high requirements for model instruction-following ability, they pose a severe challenge to open-source models. As shown in Table 3, Qwen-2.5-VL-7B could hardly complete any tasks before training (scoring only 0.9 in logistics and 0 in after-sales tasks). In fact, we further test the performance of Qwen-2.5-VL-32B[1]. Additionally, following your suggestion, experiments with two specialized e-commerce dialogue models are added. We test these models with ReAct as the framework, and the specific results are shown in the table below, where “-” indicates that the model does not support the relevant modality.
>
> | Model                | Logistics | After-sales | Pre-sales | Total |
> |----------------------|-----------|-------------|-----------|-------|
> | Qwen-2.5-vl-32B[1]   | 11.1      | 7.69        | -         | -     |
> | EcomGPT[2]           | 0.00      | -           | -         | -     |
> | eCeLLM-M[3]          | 1.85      | -           | -         | -     |
>
> The results indicate that the open-source models underperform on the Mix-Ecom task, prompting us to shift our evaluation focus to the currently leading proprietary VLLMs.
>
> # W5: No enough analysis of how fine-tuning (e.g., on Mix-ECom subsets) impacts performance, which may limit understanding of the benchmark’s value.
>
> As shown in Table 3 of the paper, the improvement in the reasoning ability of the Qwen-2.5-VL 7B model through SFT is relatively limited. We believe this is mainly due to the high difficulty of the Mix-Ecom task itself (such as extremely long context lengths and complex domain rules), coupled with the relatively small model size.
> In the later revision, we plan to train on models with larger parameters and adopt a training approach that combines SFT with RL to explore more optimal fine-tuning methods.

---

> > ### Author Response · Authors · 2025-11-17
> >
> > # W6&Q1:  No comparison with specialized e-commerce dialogue models
> >
> > Thank you for your suggestion. We have added experiments with the following e-commerce dialogue models. EcomGPT 7B and eCeLLM-M 7B are utilized as backbones and experiments are conducted with ReAct as the framework. Since these two models only support the text modality, experiments are only performed on the logistics subset. The results are as follows:
> > | Model         | Key Answer Score | Database score | Total |
> > |---------------|------------------|----------------|-------|
> > | EcomGPT[2]    | 0.00             | 45.4           | 0.00  |
> > | eCeLLM-M[3]   | 2.78             | 46.3           | 1.85  |
> >
> > Once again, the experimental results demonstrate the challenging nature of Mix-Ecom. Our benchmark not only requires models to be familiar with e-commerce business logic but also demands that models possess strong capabilities in instruction following, tool calling, and reasoning.
> >
> > # W7: No manual evaluation in the experiment part.
> >
> > In lines 434 to 427 of the paper, as well as in Table 4, we present the results of our human evaluation. Specifically, to conduct manual evaluation, we collaborate with an e-commerce company that has a professional data annotation team, whose members possess extensive experience in e-commerce customer service.
> > As shown in Table 4 of the paper, the annotation team evaluates performance based on the following three aspects:
> > 1. Human-likeness: This assesses the authenticity of the model's responses in an e-commerce customer service context. Specifically, it examines whether the wording and tone resemble that of a human customer service representative, and whether the response logic aligns with common practices in the e-commerce industry. For example, if a user requests a 5-yuan red packet compensation and the model, after checking through its tools that the merchant can compensate up to 10 yuan, directly offers 10 yuan, it violates the basic e-commerce principle of “compensating as needed and controlling costs.”
> > 2. Informativeness: Without providing the key answer content, the annotators judge whether the model's output includes all the information that users care about.
> > 3. Key answer: With the key answer content provided, the annotators determine whether the model's output includes all the information contained in the key answer.
> >
> > # Q2: Do you intend to add a user study
> > 1. As stated in W7, all members of our human annotation team have extensive experience in e-commerce customer service. Therefore, the human evaluation results shown in Table 4 of the paper can effectively reflect user satisfaction.
> > 2. In the next version, we will implement this paradigm in the customer service system of our partner company to conduct A/B testing. We will continuously monitor key metrics such as problem resolution rate, satisfaction, transfer-to-human rate, and order conversion rate.
> >
> > **Reference:**
> >
> > [1]Jinze Bai, et al. Qwen-VL: A Versatile Vision-Language Model for Understanding, Localization, Text Reading, and Beyon
> >
> > [2] Li Yangning, et al. EcomGPT: Instruction-tuning Large Language Models with Chain-of-Task Tasks for E-commerce
> >
> > [3] Bo Peng, et al. eCeLLM: Generalizing Large Language Models for E-commerce from Large-scale, High-quality Instruction Data

---

> ### Comment · Reviewer_DZnq · 2025-11-21
>
> This rebuttal has addressed many concerns and corrected my negligences.
>
> I will raise my recommendation to 6. But honestly, I prefer a score of '5'.

---

### Official Review · Reviewer_j6Yo · 2025-10-31

**Soundness:** 3
**Presentation:** 3
**Contribution:** 4
**Rating:** 6
**Confidence:** 4

**Summary:**

This paper proposes Mix-ECom, a dataset made to test chatbots used in e-commerce. It has around 4800 real-life customer chats, taken from real e-commerce cases. These chats cover different types of dialogues like question-answer, product advice, help with tasks, and small talk. It also covers different shopping stages like before buying, delivery, and after-sales. There are also some small rules about returns, shipping, refunds, and so on. Each chat can include pictures or short videos, tools like "calculate shipping cost" or "change delivery address," and info from a logistics database. The authors propose a fix to the issue of rule hallucination, when chatbots ignore or mess up company rules. For this, they make a new system called E-ReAct and E-Plan & Solve, which helps the bot focus only on the right rules at the right time. When they tested big AI models like GPT-4o and Gemini-2.5-Pro, none did great but after fine-tuning the models on Mix-ECom, performance improved.

**Strengths:**

- The dataset is of high quality: it contains real human chats, is multimodal (as and when needed), covers different aspects of conversation like casual, complaint, requests, etc. The data has rules which helps test chatbots on reasoning rather than just engagement.
- I like the testing design, where the authors separate how corrext the answer are from how well the system handles the internal state. The error breakdown of rule mistake, image misreading, or  early reaction, is clear and helpful.
- The paper is well written with a comprehensive appendix that contains almost everything that is needed to reproduce the study.

**Weaknesses:**

- Most chats may switch between the chat mix in a sequential manner rather than mixing them up randomly, e.g. after sales to recommendation to QA. This might make the chats too restrictive (more like multi-step rather than mixed) and the pattern too well known to the chatbots.
- I think the authors use the term "hallucination" too loosely. For e.g. they call 63% of mistakes as hallucinations but many of those are just missed rules or retrieval errors, not the model making things up.
- By giving the model full reasoning cot steps during training may help the model too much. Real bots don’t get these ready-made steps, so the test might look easier than it really is.
- Human reviewers checked if responses were natural and informative, but not if they followed company rules. A chatbot can sound polite and smart but still break return or refund policies and the paper should measure this too.

**Questions:**

- When you say "mixed-type", do you mean many things happening in the same line (like sarcasm + refund + product question together), or just one after another in some predefined sequence? How will you check real mixing, not just switching?
- In your dynamic system, how do you decide which rules matter? Simple keywords, embedding similarity, or a trained scoring model?
- After fine-tuning, are models memorising rule lines instead of reasoning? For example, do they just learn "mangosteen means refund only" without knowing the reason?
- Can Mix-ECom do user modelling? Like changing tone if the user is angry. You already have mood labels, so can you use them to enforce policies in a personalized way?

---

> ### Author Response · Authors · 2025-11-17
>
> # W1: Most chats may switch between the chat mix in a sequential manner rather than mixing them up randomly.
>
> Our data is entirely sourced from real e-commerce customer service conversations. During the annotation process, we only added reasoning and tool call processes and desensitized privacy information, without altering the original task types. Therefore, the combination and distribution of task types in the dataset can highly reflect the real problem composition in e-commerce scenarios. In contrast, random combination of task types would lead to a significant deviation of data distribution from the reality.
>
> # W2: I think the authors use the term "hallucination" too loosely. For e.g. they call 63% of mistakes as hallucinations but many of those are just missed rules or retrieval errors, not the model making things up.
>
> In Section 5.5 BASE CASE ANALYSIS of the paper, we analyze the assistant agent that uses GPT-4o as its backbone and ReAct as its framework. Since we do not introduce the dynamic module, we provide the complete domain rules in the prompt, so there is no issues of rule omission or retrieval errors.
> As stated in lines 458 to 465 of the paper, excessively long domain rules place extremely high demands on the model's ability to follow instructions. Such errors mainly arise from the large model's misinterpretation of the rules. These errors can be categorized as hallucinations or instruction-following mistakes. Besides, the results show the difficulty of the dataset.
> We apologize for any confusion our wording may have caused you. We will optimize the relevant expressions in subsequent versions.
>
> # W3: By giving the model full reasoning cot steps during training may help the model too much.
>
> During training, we do not provide the model with complete reasoning steps as input. As described in Section 5.3 of the paper, for a training sample that involves t reasoning steps, we break it down into t training data points. In the i-th data point, the input includes the reasoning process up to the (i-1)-th step, while the output corresponds to the content of the i-th reasoning step. This approach enables the model to gradually learn reasoning skills.
> In fact, this training method is consistent with the reasoning process that the model undergoes during evaluation or practical application. In real-world scenarios, the model first generates the first reasoning step based on the user's initial query. It then progressively advances to the next reasoning step and tool call based on the previously generated reasoning history, until the task is ultimately completed.
>
> # W4: Human reviewers checked if responses were natural and informative, but not if they followed company rules.
>
> We have collaborated with an e-commerce company that has a professional data annotation team, whose members possess extensive experience in e-commerce customer service.
> As shown in Table 4 of the paper,  the annotation team evaluates performance based on the following three aspects:
> 1. Human-likeness: This assesses the authenticity of the model's responses in an e-commerce customer service context. Specifically, it examines whether the wording and tone resemble that of a human customer service representative, and whether the response logic aligns with common practices in the e-commerce industry. For example, if a user requests a 5-yuan red packet compensation and the model, after checking through its tools that the merchant can compensate up to 10 yuan, directly offers 10 yuan, it violates the basic e-commerce principle of “compensating as needed and controlling costs.”
> 2. Informativeness: Without providing the key answer content, the annotators judge whether the model's output includes all the information that users care about.
> 3. Key answer: With the key answer content provided, the annotators determine whether the model's output includes all the information contained in the key answer.

---

> > ### Author Response · Authors · 2025-11-17
> >
> > # Q1: When you say "mixed-type", do you mean many things happening in the same line (like sarcasm + refund + product question together), or just one after another in some predefined sequence?
> >
> >
> > As shown in Figure 1 of the paper, the term "mixed-type" refers to the fact that our tasks consist of the following four types: Task-oriented Dialogue, Recommendation, Question & Answer, and Chitchat. These four types of tasks have been clearly defined in traditional NLP dialogue systems.
> > Previous e-commerce systems or benchmark studies  focus on only one of these task types. However, real-world e-commerce scenarios are actually a mixture of all four types of tasks. The specific types you mentioned can be mapped with the above categories: sarcasm corresponds to Chitchat, refund corresponds to Task-oriented Dialogue, and product question corresponds to Question & Answer.
> > Our work is the first to propose mixed-type e-commerce dialogues and create a benchmark that fully covers all e-commerce customer service scenarios.
> >
> > # Q2：In your dynamic system, how do you decide which rules matter?
> >
> > The selection of domain rules is autonomously decided by the model, without the need for additional training. In the dynamic module, the model takes into account the historical conversation records, the current reasoning trajectory, and the entire content of the domain rules. It then outputs a list of domain rule titles relevant to the current task, such as “['Basic Guidelines', 'Shipping Cost Calculation Guidelines']”. Subsequently, the system replaces the corresponding domain rules section in the prompt based on this list, achieving dynamic adjustment of the domain rules.
> >
> > # Q3: After fine-tuning, are models memorising rule lines instead of reasoning?
> >
> > As shown in Table 3 of the paper, the improvement in the reasoning ability of the Qwen-2.5-VL 7B model through SFT is relatively limited. We believe this is mainly due to the high difficulty of the Mix-Ecom task itself (such as extremely long context lengths and complex domain rules).
> > Furthermore, we plan to train on models with larger parameters and adopt a training approach that combines SFT with RL to explore more optimal fine-tuning pathways.
> >
> > # Q4: Can Mix-ECom do user modelling?Like changing tone if the user is angry. You already have mood labels, so can you use them to enforce policies in a personalized way?
> >
> > In Mix-Ecom, the user profile is a form of user modeling, which includes basic user information (such as mood, address, etc.) and current needs. We use the user profile as an instruction to enable the user agent to simulate user behavior.
> > Thanks to DeepSeek-R1's excellent imitation ability, the user agent in Mix-Ecom can highly reproduce the tone characteristics of real users. The model can not only adjust the tone based on mood labels but also flexibly use local dialects and expressions in combination with the address information in the user profile. For example, when the user's address is shown as Hong Kong or Taiwan, the user agent reply in Traditional Chinese; if the address is in other regions of China, it respond in Simplified Chinese.

---

### Official Review · Reviewer_um8L · 2025-10-31

**Soundness:** 3
**Presentation:** 2
**Contribution:** 3
**Rating:** 8
**Confidence:** 4

**Summary:**

The authors focus on the problem domain of conversational assistants for the e-commerce domain. The authors argue that current benchmarks are not comprehensive enough for real-world dialogs and come up with their own dataset called Mix-ECom. The datasets is constructed by taking real-world customer dialogs and using LLMs to remove user privacy, add reasoning between responses and simulate user profiles.

They then evaluate a series of models on this dataset and leverage the ReAcT and Plan&Solve framework for their models. They adapt those frameworks and have a module that filters out certain turns that may result in hallucination errors and show that their new setup is better. They also show finetuning on their dataset improves an open-source model.

**Strengths:**

1) The construction of a dataset that mimics real-world conversations is a good contribution to the community.

2) The set of models in the paper along with the experiments done seem comprehensive and the analysis as to why GPT-4o is much worse than Gemini 2.5 pro is an interesting contribution. The dynamic module approach also seems promising and can be explored in future work.

**Weaknesses:**

There are a couple clarifications needed in the paper

1) Who came up with these domain rules? It seems important but it's not mentioned why these are required.

2) There is alot of work on human verification but there arent specific details on the instructions given to the humans. Additionally what constitutes a high quality conversation?

3) It is mentioned that the dialog is rewritten. What does that mean? I assumed only the reasoning chain was added but it seems more was added.

4) Why are other task-oriented metrics like number of turns not evaluated.

**Questions:**

Questions

1) How did you split the dataset into train / test? Was it random or did you take into account actions that may be in the training data but aren't in the test.

2) Why is the use of the dynamic module only when the action talk_to_user outputted? Would be good to explain this.

Suggestions

1) I think in Line 018 you meant to say "multiple"

---

> ### Author Response · Authors · 2025-11-17
>
> # W1: Who came up with these domain rules
>
> Our team has partnered with a well-known e-commerce company that boasts a user base of over a hundred million. Due to the anonymity policy, we are unable to disclose its specific name. The domain rules we use are formulated based on the fundamental business logic of this company’s e-commerce operations.
>
> # W2: About human verification
>
> Our team has collaborated with a renowned e-commerce company, which has a specialized data annotation team. Members of this team possess extensive experience in e-commerce customer service.
> As shown in Table 4 of the paper, the annotation team evaluates performances based on the following three aspects:
> 1. Human-likeness: This assesses the authenticity of the model's responses in an e-commerce customer service context. Specifically, it examines whether the wording and tone resemble that of a human customer service representative, and whether the response logic aligns with common practices in the e-commerce industry. For example, if a user requests a 5-yuan compensation, and the model, after checking through its tools that the merchant can compensate up to 10 yuan, directly offers 10 yuan, it violates the basic e-commerce principle of “compensating as needed and controlling costs.”
> 2. Informativeness: Without providing the key answer content, the annotators judge whether the model's output includes all the information that users care about.
> 3. Key answer: With the key answer content provided, the annotators determine whether the model's output includes all the information contained in the key answer.
>
> We will supplement the above content in subsequent versions.
>
> # W3: It is mentioned that the dialog is rewritten. What does that mean?
>
> As shown in Figure 2, during the data annotation phase, since the original conversational data do not include tool calls or reasoning processes, reasoning chain processes are added to the dialogues for further model training. Additionally, due to that the dialogues may involve sensitive information such as the names, addresses, and phone numbers of users and merchants, privacy processing is conducted. We use LLM to uniformly delete or replace these fields to ensure data privacy and security.
>
> # W4: Why are other task-oriented metrics like number of turns not evaluated.
>
> The user behavior in mix-ecom may be dynamicly changing during dialogues, leading to the increase of the number of dialogue turns. It needs to be emphasized that these increase in dialogue turns have no bearing on the actual performance of the assistant agent.
> Moreover, considering that in real e-commerce scenarios, platforms focus more on the resolution rate of issues rather than the number of dialogue turns, we have identified the key metrics representing resolution rate—key answer score and database score—as the evaluation criteria for mix-ecom benchmark.
>
> # Q1: How did you split the dataset into train / test?
> The training set and test set are randomly divided and share the same distribution, both comprehensively covering all types of issues in e-commerce scenarios. Additionally, each instance of data originates from a different user, ensuring strict separation between the training set and test set in terms of data sources.
>
> # Q2:Why is the use of the dynamic module only when the action talk_to_user outputted？
>
> The design of our Dynamic Module is motivated by observations of user behavior in e-commerce dialogue scenarios: user needs are not static, but rather evolve dynamically throughout the conversation. Therefore, the system must be capable of dynamically selecting relevant Domain Rules during dialogue reasoning and flexibly adjusting its planned path.
>
> The Dynamic Module is triggered when a user initiates their first query, or when the assistant agent receives a new user query through the talk_to_user tool. Upon activation, the module re-filters relevant domain rules and generates a new task plan.

---

### Author Response · Authors · 2025-11-17

# Response to All Reviewers
We would like to thank all the reviewers for their insightful reviews and valuable suggestions. We are excited that  [Reviewer um8L, Reviewer j6Yo, Reviewer DZnq, Reviewer bcjS] think our work is a good contribution to the community, [Reviewer j6Yo] think the dataset is well-designed, and [Reviewer um8L, Reviewer j6Yo, Reviewer DZnq] think the experiments are comprehensive and clear.
We also appreciate the suggestions provided. We have noticed that several reviewers have raised concerns regarding the legitimacy of the original dialogue sources and user profile information used in our data construction. Here, the following clarification is provided:

1. This work is conducted in collaboration with a well-known e-commerce company. Due to anonymity requirements, we are unable to disclose the name of the company. However, we can state that the company has hundreds of millions of users and employs a professional data annotation team, whose members all hvae extensive knowledge of e-commerce customer service scenarios;
2. The original dialogue data used is derived from high-quality conversations generated in the company's actual business operations;
3. The domain rules are constructed based on the company's real business logic;
4. Regarding privacy protection, as shown in steps five and six in Figure 2 of the paper,  rewriting by LLMs and manual screening are applied;
5. In addition, the dataset have undergone joint review by the security and legal departments of the partner company and will be publicly available.


# Supplementary experiments
1. Experiments with GPT4o as the backbone and ReAct as the framework using the 2-shot setting.
2. Experimental results of Qwen-2.5-32B-VL[1] and two e-commerce LLMs: EcomGPT 7B[2] and eCeLLM-M[3].
3. Distribution statistics of Domain Rules used in the evaluation data.
4. Distribution statistics of task types in the evaluation data.

Reference:

[1] Jinze Bai, et al. Qwen-VL: A Versatile Vision-Language Model for Understanding, Localization, Text Reading, and Beyon

[2] Li Yangning, et al. EcomGPT: Instruction-tuning Large Language Models with Chain-of-Task Tasks for E-commerce

[3] Bo Peng, et al. eCeLLM: Generalizing Large Language Models for E-commerce from Large-scale, High-quality Instruction Data

---

### Meta-Review · Area_Chair_iqnE · 2026-01-07

**Summary:**

This work addresses the challenge of handling mixed-type dialogues and complex domain rules in LLM-based e-commerce agent benchmarks. To this end, the authors introduce Mix-ECom, a dataset derived from 70,000 real-world customer-service dialogues, ultimately retaining 4,799 samples. Mix-ECom features four dialogue types (QA, recommendation, task-oriented dialogue, chit-chat), three e-commerce task categories (pre-sales, logistics, after-sales), and incorporates 82 domain rules.

The dataset construction, which reflects real-world conversations, is a valuable contribution. The proposed dynamic module approach is also promising for future exploration.

I recommend for acceptance.

**Reviewer Concerns:**

The authors addressed all concerns

**Reviewer Scores:**

N/A

---

### Decision · Program_Chairs · 2026-01-26

Accept (Poster)